# Inhibitory Activity of Flavonoids, Chrysoeriol and Luteolin-7-*O*-Glucopyranoside, on Soluble Epoxide Hydrolase from *Capsicum chinense*

**DOI:** 10.3390/biom10020180

**Published:** 2020-01-24

**Authors:** Jang Hoon Kim, Chang Hyun Jin

**Affiliations:** Advanced Radiation Technology Institute, Korea Atomic Energy Research Institute, Jeongeup, Jeollabuk-do 56212, Korea; oasis5325@gmail.com

**Keywords:** flavonoids, soluble epoxide hydrolase, non-competitive mode, induced fit, lock-and-key

## Abstract

Three flavonoids derived from the leaves of *Capsicum chinense* Jacq. were identified as chrysoeriol (**1**), luteolin-7-*O*-glucopyranoside (**2**), and isorhamnetin-7-*O*-glucopyranoside (**3**). They had IC_50_ values of 11.6±2.9, 14.4±1.5, and 42.7±3.5 µg/mL against soluble epoxide hydrolase (sEH), respectively. The three inhibitors (**1**–**3**) were found to non-competitively bind into the allosteric site of the enzyme with *K*_i_ values of 10.5 ± 3.2, 11.9 ± 2.8 and 38.0 ± 4.1 µg/mL, respectively. The potential inhibitors **1** and **2** were located at the left edge ofa U-tube shape that contained the enzyme active site. Additionally, we observed changes in several factors involved in the binding of these complexes under 300 K and 1 bar. Finally, it was confirmed that each inhibitor, **1** and **2**, could be complexed with sEH by the “induced fit” and “lock-and-key” models.

## 1. Introduction

Arachidonic acid is converted to epoxyeicosatrienoic acids (EETs) by cytochrome P450 epoxygenase [1]. EETs exist as four regioisomeric metabolites; 5,6-, 8,9-, 11,12- and 14,15-EETs [1]. Furthermore, soluble epoxide hydrolase (sEH, E.C.3.3.2.10) biosynthesizes dihydroxyeicosatrienic acids (DHETs) from EETs by hydrolyzing the epoxide ring of EETs into a diol [2]. Since EETs were revealed as endothelium-derived hyperpolarizing factors in 1996, they have been found to have biologicalactivities, such as anti-inflammatory, anti-hypertensive, kidney-protective, and cardiac-protective effects [3]. 11,12-EET was reported to inhibit nuclear factor kappa B signaling pathways and cytokine-induced expression of cellular adhesion molecules [4]. 11,12- and 14,15-EETs stimulate the large-conductance Ca^2+^-activated K^+^ channel and dilate blood vessels [3]. Therefore, sEH, which is encoded in the *EPHX2* gene located at chromosomal region 8p21-p12, has been known as the hydrolase to treat cardiovascular disease [5]. The sEH inhibitor, 12-(3-adamantan -1-yl-ureido)dodecanoic acid (AUDA), is a potential inhibitor, with an IC_50_ value in the nanomolar range [6]. Due to its low solubility, AUDA is dissolved in DMSO for in vitro experiments and used as a formulation with large amounts of 2-hydroxylpropyl β-cyclodextrin in vivo [3].

There are 30 species in the genus *Capsicum*, and of them *Capsicum annuum* L. (*C. annuum*), *Capsicum chinense* Jacq. (*C. chinense*), *Capsicum frutescens* L., *Capsicum baccatum* L., and *Capsicum pubescens* Ruiz and Pav. are mainly cultivated [7]. *C.*
*chinense* or “Bhoot jolokia” and “Bin jolokia” in Assam, “Naga King Chili” in Nagaland, “Umorok” in Manipur, and “Ghost pepper” in the west, which belongs to the family Solanaceace [8], is an important horticultural crop for food and medicines [9]. This fruit has been reported to be one of the hottest chili peppers [10]. Capsaicin analogs, nornordihydrocapsaicin, nordihydrocapsaicin, capsaicin, dihydrocapsaicin, homocapsaicin, and homodihydrocapsacin, were detected in the extract of the fruit of *C. chinense* [11]. Additionally, several flavonoids have been reported to have been isolated from the genus *Capsicum* [12]. The content of quercetin in the leaves is higher in the immature pepper stage (~156.9 mg/g) than in the mature pepper stage (~10.2 mg/g) [12]. Recently, flavonoids, kaempferol and apigenin from *Tetrastigma*
*hemsleyanum* [13], and the flavonoid glycosides, quercetin-3-*O*-*β*-D-galactoside and 3,7,3′-tri-*O*-methylquercetin-4′-*O*-*β*-D-apiofuranosyl-(1→2)-*O*-*β*-D-glucopyranoside [14], have been reported as non-competitive agents blocking the catalytic reaction of sEH [13,14]. Finally, these findings led to the analysis of the components from leaves of *C. chinense* and study of their activity against sEH.

## 2. Materials and Methods

### 2.1. General Experimental Procedures

Column chromatography was performed using silica gel (Kieselgel 60, 70–230 and 230–400 mesh, Merck, Darmstadt, Germany), Sephadex LH-20 (GE Healthcare, Uppsala, Sweden), and C-18 (ODS-A 12 nm S-150, S-75 μm; YMC Co., Kyoto, Kansai, Japan) resins. Thin-layer chromatography (TLC) was performed using pre-coated silica gel 60 F254 and RP-18 F254S plates (both 0.25 mm, Merck). Spots in the TLC were visualized by spraying with 10% aqueous H_2_SO_4_ solution followed by heating to 300℃ in dried air. Nuclear magnetic resonance (NMR) spectra were recorded using the JEOL ECA 500 spectrometer (^1^H, 500 MHz; ^13^C, 125 MHz) (JEOL, Tokyo, Japan) (Appendix A). AUDA (10007927), soluble epoxide hydrolase (10011669) and PHOME (10009134) were purchased from Cayman (Cayman, Ann Arbor, MI, USA).

### 2.2. Plant Materials

The leaves of *C. chinense* were collected in October 2017 at Jeollbuk-do, Republic of Korea, and were identified by Dr. Y.D. Jo in the Radiation Breeding Research Center (RBRC), Korea Atomic Energy Research Institute (KAERI). A sample specimen (RBRC002) was deposited at the Herbarium of RBRC, KAERI, Republic of Korea. 

### 2.3. Extraction and Isolation 

The leaves of *C. chinense* (2 kg) were extracted twice with 95% methanol (36 L) at room temperature for a week. The solution was evaporated under reduced pressure to obtain a methanol extract (~210 g). The brown extract was suspended in distilled water (2.1 L) and successively partitioned with *n*-hexane, chloroform, ethyl acetate, and BuOH fractions. The chloroform fraction (8.4 g) was chromatographed on a silica gel column and eluted with a gradient system of *n*-hexane-acetone (0:1→1:1) to give eight fractions (HC1–HC8). The HC6 fraction was re-chromatographed on the C-18 column using an isocratic system of methanol–water (3:1) to afford compound **1** (12 mg). The ethyl acetate fraction (6.4 g) was separated on a silica gel column eluted with a gradient solvent system of chloroform–methanol (0:1→1:2) to achieve ten fractions (HE1–HE10). The HE6 fraction was separated on a Sephadex LH-20 column with 95% methanol to obtain compound **2** (20 mg) and five fractions (HE61–HE65). Compound **3** (7 mg) was purified by C-18 column chromatography with an isocratic solvent system of 50% methanol from the HE64 fraction.

### 2.4. sEH Assay and Kinetic Analysis

sEH (130 µL, ~16 ng/mL) in 25 mM bis-Tris-HCl buffer (pH 7.0) containing 0.1% BSA and 20 µL of compound in MeOH was mixed in a 96-well plate, followed by the addition of 50 µL buffer containing 5 µM PHOME. The reactions were carried out at 37°C for 40 min. The inhibitory ratio was calculated according to the following equations: Inhibitory activityrate (%) = [(ΔC/ΔS)/ΔC] × 100(1)
where ΔC andΔS represent the intensity of control and inhibitor after 40 min, respectively.
y = y_0_ + [(a × x)/(b + x)](2)
where y_0_ is the minimum value on the y-axis, a denotes the difference between maximum and minimum values, and b refers to the x value at 50%.

### 2.5. Docking Study of sEH With Inhibitor

The docking study of sEH with its inhibitor was performed using Autodock version 4.2. The 3D structure of the inhibitor was developed and minimized by MM2. The single bonds were confirmed as flexible bonds using the AutoDockTools. The 3D structure of the protein was downloaded from the RCSB protein data bank (pdb id: 3ANS). Water, substrate, and the B chain were deleted from the pdb file. Hydrogen atoms and Gasteiger charges were added to the A chain. The grid dimensions were set at 126×126×126 number of points (x×y×z). Docking simulation was performed to dock the inhibitor 25,000,000 times into this grid. These results were visualized using Chimera version 1.14 (San Francisco, CA, USA) and AutoDockTools 1.5.6 (La Jolla, CA, USA).

### 2.6. Molecular Dynamic Study

A molecular dynamic study was performed to simulate the optimal configuration of sEH with the flavonoids using the Gromacs version 4.6.5. Their Itp and gro files were constructed in the GlycoBioChem PRODRG2 server, and integrated into gro via Gromos96 43a1 force field and topology files of sEH that were generated by utilizing gdb2gmx of Gromacs. The complex was obtained as cubes of water molecules measuring 8.5×8.5×8.5 containing six chloride ions. The energy of the complex was lowered to 10 kJ/mol of maximal force in the steepest descent method. A stable simulated complex was equilibrated at 300 K for 100 ps, and simulated at 1 bar for 100 ps. Finally, the complex was simulated for molecular dynamics for 10 ns.

### 2.7. Data Analysis

Data are expressed as the means ± standard deviation (*n* = 3). Allvalues were analyzed using Sigm aPlot (Systat Software Inc., San Jose, CA, USA) to determine treatment variations.

## 3. Results and Discussion

### 3.1. Isolation, Identification, and Enzyme Assay

Flavonoids and polyphenols were detected in chili peppers by a total flavonoid and polyphenol content test [15]. Previous phytochemical studies have reported that flavonoids, such as luteolin and quercetin derivatives, are contained in the components of hot pepper fruit [16]. In pepper leaves, 10 carotenoids were analyzed by comparing the chromatogram retention time of a mixed standard of carotenoids by using the C_30_ column of high performance liquid chromatography [17]. Based on these facts, the objective of this study was to find flavonoid derivatives from the leaves of *C. chinense*.

The dried leaves of *C. chinense* were collected from an experimental farm and extracted with 95% methanol at room temperature for a week. The concentrated extracts were dissolved in distilled water and successively divided in *n*-hexane, chloroform, ethyl acetate, and BuOH fractions. Flavonoid aglocone (**1**) and glycosides (**2** and **3**) were purified with open column chromatography from chloroform and ethyl acetate fractions, respectively. Their chemical structures were evaluated as chrysoeriol (**1**) [18], luteolin-7-*O*-glucopyranoside (**2**) [19], and isorhamnetin-7-*O*-glucopyranoside (**3**) [20] from nuclear magnetic resonance analysis, and by comparison with reported data (Figure 1).

Recently, our studies have revealed that some flavonoids showed inhibitory activity against sEH [13,14]. Purified compounds **1**–**3** were evaluated for binding strength and type of enzyme in vitrofor insight into the sEH inhibition. The inhibitory activity rate on sEH at each concentration of the compound was calculated as the product difference between the presence and absence of the enzyme using Equation (1). Figure 2A shows that they inhibit sEH activity in a dose-dependent manner at concentrations of ~1–30 µg/mL. To calculate the half maximal inhibitory concentration of inhibitors (IC_50_), these inhibition rate values were fitted by Equation (2). Inhibitors **1**–**3** were found to have IC_50_ values of 11.6 ± 2.9, 14.4 ± 1.5, and 42.7 ± 3.5 µg/mL, respectively (Table 1). 

Additionally, these inhibitors were evaluated at three different concentrations for the initial velocity (*v*_0_) at various concentrations of the substrate (1, 1.5, 2.2, 3.1, 6.2, and 12.5 µM). The inverse of these concentrations and *v*_0_ values were used for the Lineweaver–Burk plot. As indicted in Figure 2B–D, the respective concentrations of the inhibitor had family lines calculated. Flavonoids **1**–**3** exhibited the same non-competitive mechanism of actionin the enzyme with a −1/*K*_m_ value and different 1/*V*_max_ values (Table 1). Furthermore, the Dixon plot is represented by the modified Lineweaver–Burk plot. The *K*_i_ values were calculated as 10.5 ± 3.2, 11.9 ± 2.8 and 38.0 ± 4.1 µg/mL, respectively (Figure 2E–G, Table 1).

### 3.2. Molecular Docking

In 1894, the “lock-and-key” model was first suggested as the interaction of enzyme (lock) with substrate (key) by Emil Fisher [21]. Molecular docking is a virtual technology that calculates this model using Force-field [22]. Since the 1980s, computer technologies have been used for drug development by Merck Inc. [22]. Our study adopted molecular docking using the Autodock 4.2 program for increasing the understanding of the flavonoid–sEH interaction. Based on the enzyme kinetic results, we performed the blind docking to search for an allosteric site to which inhibitors **1** and **2** with IC_50_ values below 50 µg/mL could bind. We found that two had the best pose into sEH with binding energies of −8.4 and −7.6 kcal/mol, respectively (Figure 3A, Table 2). They were docked into the left edge position of a U-tube shape containing the catalytic site of the enzyme. As shown in Figure 3A–E and Table 2, flavonoid aglycone **1** had three hydrogen bonds between a ketone group and Trp525 (at a distance of 2.2 Å), a hydroxyl group and Asp496 (at a distance of 1.7 Å) or Phe497 (at a distance of 2.0 Å). The respective A- and B-rings of the flavonoid kept the π-π interactions with Phe497 at distances of 6.7 and 10.9 Å. The flavonoid glycoside (**2**) formed four hydrogen bonds between the sugar group and Leu417 and Met419 at distances of 1.9 and 2.0 Å, as well as two hydroxyl groups between the B-ring and His 524 at 1.9 Å and 2.1 Å, respectively. In addition, the B-ring of **2** kept π-π interactions with the furan ring of His420 at a distance of 12.9 Å.

### 3.3. Molecular Dynamics 

Since the bovine pancreatic trypsin inhibitor in a vacuum was simulated using molecular dynamics (MD) by McCammon in 1977, MD has been one of the state-of-the-art computational techniques for in-depth understanding of the communication between ligand and receptor with structural changes over time [22]. An induced fit model of enzyme–inhibitor binding was suggested by Koshl and because the inhibitor can induce the conformation change of the enzyme during the binding [22]. NMR and X-ray crystallography have provided limited information of the static binding pose of a ligand in an enzyme [23]. This study performed MD simulation using the Groamcs 4.6.5 program and the induced fit model to solve the unveiled communication of flavonoids (**1** and **2**) with sEH.

As shown in Figure 4A,B, inhibitors **1** and **2** visually confirmed the stable state fitted with sEH during the simulation time with ~9.1×10^5^ kJ/mol potential energy (Figure 4C). The protein root-mean-square deviation (RMSD) values of **1** and **2** were ~2.5 and ~2.8 Å, respectively (Figure 4D). The sugar group of **2** hung on the narrow entrance of the U-tube shape in sEH. Whereas **1** was located in the inner pocket of it, which was of sufficient size. The loops around the enzyme that were bound with the former produced more apparent changes than those ofthe latter (Figure 4A,B). Inhibitors **1** and **2** maintained 0–3 hydrogen bonds with residues of sEH for 10 ns (Figure 4E,F). Hydrogen bonds in the complex of sEH with inhibitors were analyzed per 1-ns interval during their trajectory (Figure 4G,H, Table 3). Noticeably, the ketone and 5-hydroxyl groups of **1** were continually maintained at a distance of ~3.5 Å from Arg410 (Figure 4G). The ketone and 6′-hydroxyl groups in the sugar of **2** remained at a distance of 3.5 Å from Met188 and Leu186 within 5 ns. The 1′-hydroxyl group in the sugar of **2** approached 3.5 Å for hydrogen bonds for 9–10 ns (Figure 4H). Based on the results in comparison to the hydrogen bonds in molecular docking and dynamics, it was found that a new hydrogen bond to Arg410 mainly occurs during dynamics in the inhibitor **1**–sEH complex. The inhibitor **2**–sEH complex constantly maintained hydrogen bonds with Leu417 and Met419 in the docking results. Therefore, inhibitors **1** and **2** were close to the “induced fit” and “lock-and-key” models, respectively.

## 4. Conclusions

Among flavonoids **1**–**3** from the dried leaves of *C. chinense*, chrysoeriol (**1**) and luteolin-7-*O*-glucopyranoside (**2**) showed inhibitory activities against sEH with IC_50_ values of 11.6 ± 2.9 and 14.4 ± 1.5 µg/mL, respectively. Through enzyme kinetic study, Lineweaver–Burk and Dixon plots revealed that they were non-competitive inhibitors with *K*_i_ values of 10.5 ± 3.2 and 11.9 ± 2.8 µg/mL, respectively. They were visually represented to be bound into the right edge, next to the catalytic site of the enzyme, by the stable Autodock score. In particular, inhibitor **1** was found to form a hydrogen bond with another amino acid (Arg410) in the molecular docking result via dynamic, whereas **2** was confirmed to form hydrogen bonds with common amino acids (Leu417 and Met419) between the molecular docking and dynamics results. These findings confirmed the possibility that the leaves of *C. chinense* containing flavonoids (**1** and **2**) were natural sEH inhibitors in vitro and in silico. Finally, the study suggests that the two are suitable for cell-based and in vivo experiments involving cardiovascular disease.

## Figures and Tables

**Figure 1 biomolecules-10-00180-f001:**
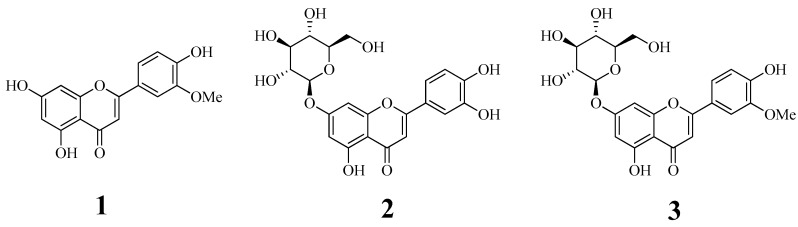
The structure of flavonoid derivatives (**1**–**3**).

**Figure 2 biomolecules-10-00180-f002:**
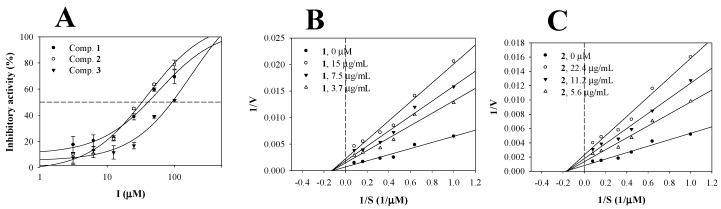
Inhibitory activities of inhibitors **1**–**3** on sEH (**A**). Lineweaver–Burk plots (**B**,**D**) and Dixon plots (**E**,**F**) for the inhibition of sEH by **1**–**3**.

**Figure 3 biomolecules-10-00180-f003:**
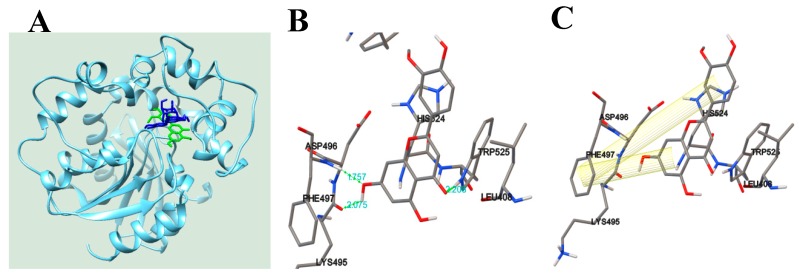
Predicted docking poses of **1** (green) and **2** (blue) into sEH (**A**). The hydrogen bond (**B**) and π-π interaction (**C**) of sEH with **1**. The hydrogen bond (**D**) and π-π interaction (**E**) between sEH and **2**.

**Figure 4 biomolecules-10-00180-f004:**
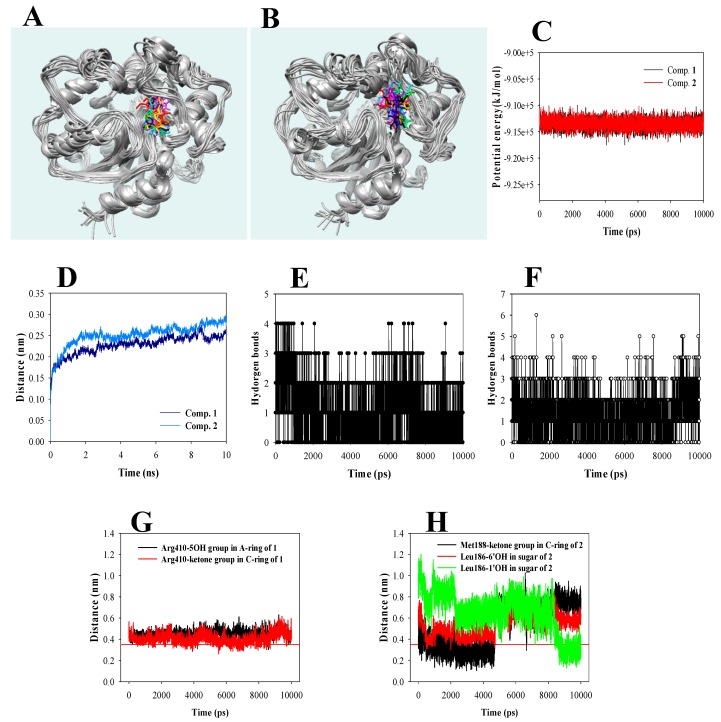
Superpositions of inhibitors **1** (**A**) and **2** (**B**) against sEH for 1 ns (red: 0, orange: 1, yellow: 2, green: 3, forest green: 4, cyan: 5, blue: 6, conflower blue: 7, purple: 8,magenta: 9, and black: 10 ns). The potential (**C**), RMSD (**D**), and hydrogen bond numbers (**E**,**F**) of **1** and **2** with sEH. The distance (**G**,**H**) of compounds **1** and **2** with key amino acids, respectively.

**Table 1 biomolecules-10-00180-t001:** The soluble epoxide hydrolase (sEH) inhibitory activities and enzyme kinetics of flavonoids.

	The Inhibitory Activity on sEH	
	IC_50_ (µg/mL) ^a^	Binding Mode (µg/mL)
**1**	11.6 ± 2.9	Non-competitive type (10.5 ± 3.2)
**2**	14.4 ± 1.5	Non-competitive type (11.9 ± 2.8)
**3**	42.7 ± 3.5	Non-competitive type (38.0 ± 4.1)
AUDA ^b^	1.2 ± 1.2 (ng/mL)	

^a^ all compounds were tested in a set of triplicated experiments. ^b^ Positive control.

**Table 2 biomolecules-10-00180-t002:** Interaction and binding energy of sEH with flavonoids (**1** and **2**).

	Hydrogen Bonds (Å)	π-π Interaction (Å)	Binding Energy (kcal/mol)
**1**	Asp496(1.7), Phe497(2.0), Trp525(2.2)	Phe497(6.7, 10.9)	−8.4
**2**	Asp335(1.9,2.1), Leu417(1.9), Met419(2.0)	His420(12.9)	−7.6

**Table 3 biomolecules-10-00180-t003:** Hydrogen bond analysis of the inhibitors with sEH at 1-ns intervals for 10 ns.

Time (ns)	1	2
	Amino Acid (Å)	Amino Acid (Å)
0	Arg410(2.78), Tyr466(2.78)	His420(2.83), Ser415(2.80), Leu417(3.32), Tyr466(3.01)
1	Arg410(2.69,2.94), Phe497(2.94)	Phe267(3.20), Leu417(2.82), Trp525(3.01)
2	Arg410(3.06,3.28), Ser412(3.35)	Arg410(3.06), Ser415(2.87), Leu417(2.98), Met419(2.80)
3	Arg410(2.95)	Ser418(3.35), Met419(3.32), His420(3.15)
4	Arg410(2.74, 3.24), Ser412(3.21)	Arg410(3.29), Met419(3.30)
5	Arg410(3.17, 3.18)	
6	Arg410(3.07)	Met419(3.08)
7	Arg410(3.16), Ser412(2.89), Phe497(3.19)	Met419(3.03)
8	Arg410(3.21,3.04), Trp525(2.86)	Phe267(3.22)
9	Arg410(2.92), Ala411(3.04)	Leu417(2.92), Phe497(2.82,2.86), Lys495(2.94)
10	Arg410(2.61,3.10), Ala411(2.77)	Leu417(2.87), His420(3.18), Ser412(3.19), His524(3.18)

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
