# Peer review of "Inhibitory Activity of Flavonoids, Chrysoeriol and Luteolin-7-O-Glucopyranoside, on Soluble Epoxide Hydrolase from Capsicum chinense"

_biomolecules, 2020, doi:10.3390/biom10020180_

Round 1
Reviewer 1 Report
Figures need to be corrected and minor spell check needed
Author Response
Reviewers' comments
Reviewer 1
The authors would like to thank Reviewer #1 for the considering our manuscript interesting and accepting it for publication after minor revision. We have incorporated the following reviewer’s specific comments in preparation of revised version of manuscript.
Figures need to be corrected and minor spell check needed
Answer: We revised.

Reviewer 2 Report
Dear Authors,
Please find the report for manuscript "Inhibitory activity of flavonoids, chrysoeriol and luteolin-7-O-glucopyranoside, on soluble epoxide hydrolase".
The title may revised as "Inhibitory activity of flavonoids, chrysoeriol and luteolin-7-O-glucopyranoside, on soluble epoxide hydrolase from Capsicum Chinense"
Line no 227: the sentence is not complete. Please check
Author Response
Reviewer 2
The authors would like to thank Reviewer #2 for the considering our manuscript interesting and accepting it for publication after minor revision. We have incorporated the following reviewer’s specific comments in preparation of revised version of manuscript.
Dear Authors,
Please find the report for manuscript "Inhibitory activity of flavonoids, chrysoeriol and luteolin-7-O-glucopyranoside, on soluble epoxide hydrolase".
The title may revised as "Inhibitory activity of flavonoids, chrysoeriol and luteolin-7-O-glucopyranoside, on soluble epoxide hydrolase from Capsicum Chinense"
Answer: We revised the title of manuscript.
Line no 227: the sentence is not complete. Please check
Answer: We revised this sentence.

Reviewer 3 Report
The manuscript presents a study of characterization and identification of three flavonoids from the leaves of Capsicum chinense Jacq. These compounds are evaluated for their inhibitory activity on soluble epoxide hydrolase and the study reports the binding site of these compounds. The methods and results are presented well, however to improve the quality of publication following points needs to be addressed-
Authors should pay attention to the English throughout the manuscript. E.g.: Line23”Interaction” instead of Introduction. Line 32 “EEts” instead of EETs. Line 42”western media”. Line 47 “Additionally, flavonoids were found the components of Capsicum [12]” is an incomplete sentence. Line 69 “voucher specimen” instead of sample specimen. Line 131” Recently, our studies” instead of other studies are a few examples. Introduction needs to explain the rationale of selecting leaves as a source of extracting the compounds. The summary of the study needs to be written clearly in the introduction and abstract. Methods section (“Extraction and Isolation”) does not explain why HC6 fraction was chosen for extraction. Similarly for other fractions. It needs to be clearly stated how it was decided that why these fractions were taken. Methods section under “sEH assay and kinetic analysis” mentions reactions were carried out for 40minutes, where in the same section it states “where ΔC and ΔS represent the intensity of control and inhibitor after 20 min”. Explain the discrepancies. Methods section “Docking study of sEH with inhibitor” , explain why Hydrogen atoms and Gasteiger charges were added to A chain. Table 1. Explain AUDA. Figures: Legends needs to be more descriptive for figure 2 and 3. This holds true for other figures in the main and supplementary documents. Quality of figures needs to be improved. The NMR plots in supplementary are not legible. Conclusions looks like the summary of results, and does not conclude w implications of this study. Also, authors should refer to other literature and draw more conclusions. Conclusion section needs to be rewritten.Author Response
Reviewer 3
The manuscript presents a study of characterization and identification of three flavonoids from the leaves of Capsicum chinense Jacq. These compounds are evaluated for their inhibitory activity on soluble epoxide hydrolase and the study reports the binding site of these compounds. The methods and results are presented well, however to improve the quality of publication following points needs to be addressed.
The authors would like to thank Reviewer #3 for the considering our manuscript interesting and accepting it for publication after minor revision. We have incorporated the following reviewer’s specific comments in preparation of revised version of manuscript.
Authors should pay attention to the English throughout the manuscript. E.g.: Line23”Interaction” instead of Introduction.
Answer: We revised this according to reviewer’s comment.
Line 32 “EEts” instead of EETs.
Answer: We revised this according to reviewer’s comment.
Line 42”western media”.
Answer: the western media
Line 47 “Additionally, flavonoids were found the components of Capsicum [12]” is an incomplete sentence.
Answer: We revised this sentence
Line 69 “voucher specimen” instead of sample specimen.
Answer: We revised this.
Line 131” Recently, our studies” instead of other studies are a few examples.
Answer: No results have been studied and reported except in our team.
Introduction needs to explain the rationale of selecting leaves as a source of extracting the compounds.
Answer: During the study of Red Pepper, the paper reported flavonoids in leaves. We previously confirmed that flavonoids have an effect on sEH. Therefore, we applied flavonoids isolated from leaves to sEH activity.
The summary of the study needs to be written clearly in the introduction and abstract.
Answer: We rewrote clearly the summary.
Methods section (“Extraction and Isolation”) does not explain why HC6 fraction was chosen for extraction.
Answer: HC6 is the main fraction from which flavonoids were isolated.
Similarly for other fractions.
It needs to be clearly stated how it was decided that why these fractions were taken.
Answer: Other fractions are the fraction from which flavonoids were isolated.
Methods section under “sEH assay and kinetic analysis” mentions reactions were carried out for 40minutes, where in the same section it states “where ΔC and ΔS represent the intensity of control and inhibitor after 20 min”. Explain the discrepancies.
Answer: It is our mistake. We confirmed a time course of 40 minutes for inhibition and 20 minutes for kinetic.
Methods section “Docking study of sEH with inhibitor” , explain why Hydrogen atoms and Gasteiger charges were added to A chain.
Answer: SEH consists of A and B chains of the same structure. The B chain was removed for reasonable experimentation. The experiment was conducted according to the program instructions.
Table 1. Explain AUDA.
Answer: AUDA is positive control.
Figures: Legends needs to be more descriptive for figure 2 and 3. This holds true for other figures in the main and supplementary documents. Quality of figures needs to be improved. The NMR plots in supplementary are not legible.
Answer: We added expended figure
Conclusions looks like the summary of results, and does not conclude w implications of this study. Also, authors should refer to other literature and draw more conclusions. Conclusion section needs to be rewritten.
Answer: For author, conclusion is the summary of this study. Discussion of this study has been already finished in results and discussion part.

Reviewer 4 Report
Kim and Jin presented inhibitory effect of flavonoids from the leaves of Capsicum chinense Jacq on soluble epoxide hydrolase (sEH). Firstly, they identified three flavonoids, chrysoeriol (1), luteolin-7-O-glucopyranoside (2), and isorhametin-7-O-glucopyranoside (3). Using enzyme kinetic study, they showed that flavonoids 1 and 2 non-competitively inhibit sEH. Moreover, they modeled the flavonoid-sEH interaction by molecular docking and molecular dynamics simulation. Flavonoids 1 and 2 were docked into left edge of U-tube shape containing the catalytic site of sEH, and were complexed with the enzyme by ‘induced fit’ and ‘lock-and-key’ models, respectively.
Broad comments
The first paragraph of introduction needs to be restructured with logic.
Researches have been showing inhibitory effect of flavonoids on sEH. What will this study contribute to the field? The results of the manuscript are convincing, but further discussion is needed.
Specific comments
As flavonoids are chemicals but not enzymes, it seems not appropriate to use activity here.
This study suggests that flavonoids 1 and 2 are natural sEH inhibitors in vitro and in silico. What are the concentrations of these two compounds in the leaves of C. chinense? Are there any other plants known to contain higher concentration of these two flavonoids?
Author Response
Reviewer 4
Kim and Jin presented inhibitory effect of flavonoids from the leaves of Capsicum chinense Jacq on soluble epoxide hydrolase (sEH). Firstly, they identified three flavonoids, chrysoeriol (1), luteolin-7-O-glucopyranoside (2), and isorhametin-7-O-glucopyranoside (3). Using enzyme kinetic study, they showed that flavonoids 1 and 2 non-competitively inhibit sEH. Moreover, they modeled the flavonoid-sEH interaction by molecular docking and molecular dynamics simulation. Flavonoids 1 and 2 were docked into left edge of U-tube shape containing the catalytic site of sEH, and were complexed with the enzyme by ‘induced fit’ and ‘lock-and-key’ models, respectively.
The authors would like to thank Reviewer #3 for the considering our manuscript interesting and accepting it for publication after minor revision. We have incorporated the following reviewer’s specific comments in preparation of revised version of manuscript.
Broad comments
The first paragraph of introduction needs to be restructured with logic.
Researches have been showing inhibitory effect of flavonoids on sEH. What will this study contribute to the field? The results of the manuscript are convincing, but further discussion is needed.
Answer: During the study of Red Pepper, the paper reported flavonoids in leaves. We previously confirmed that flavonoids have an effect on sEH. Therefore, we applied flavonoids isolated from leaves to sEH activity.
Specific comments
As flavonoids are chemicals but not enzymes, it seems not appropriate to use activity here
This study suggests that flavonoids 1 and 2 are natural sEH inhibitors in vitro and in silico. What are the concentrations of these two compounds in the leaves of C. chinense? Are there any other plants known to contain higher concentration of these two flavonoids?
Answer: The leaves of C. chinenes are waste. The study sought to verify the utility of the ingredients it contains. Furthermore, content research of compounds is another field of research.

Round 2
Reviewer 4 Report
The authors have clarified questions I raised in my previous review.